# Burns in the Elderly: Potential Role of Stem Cells

**DOI:** 10.3390/ijms21134604

**Published:** 2020-06-29

**Authors:** Margarita Elloso, Ankita Kambli, Ayesha Aijaz, Alex van de Kamp, Mark G. Jeschke

**Affiliations:** 1Sunnybrook Research Institute, Sunnybrook Health Sciences Centre, Toronto, ON M4N3M5, Canada; ankita.kambli@sri.utoronto.ca (A.K.); ayesha.aijaz@sri.utoronto.ca (A.A.); alex.vandekamp@sri.utoronto.ca (A.v.d.K.); marc.jeschke@sunnybrook.ca (M.G.J.); 2Department of Laboratory Medicine and Pathobiology, University of Toronto, Toronto, ON M5S1A8, Canada; 3Ross Tilley Burn Centre, Sunnybrook Health Sciences Centre, Toronto, ON M4N3M5, Canada; 4Institute of Medical Science, University of Toronto, Toronto, ON M5S1A8, Canada; 5Division of Plastic and Reconstructive Surgery, Department of Surgery, Faculty of Medicine, University of Toronto, Toronto, ON M5T1P5, Canada

**Keywords:** elderly, mesenchymal stem cells (MSCs), stem cells, wound healing, burn

## Abstract

Burns in the elderly continue to be a challenge despite advances in burn wound care management. Elderly burn patients continue to have poor outcomes compared to the younger population. This is secondary to changes in the quality of the aged skin, leading to impaired wound healing, aggravated immunologic and inflammatory responses, and age-related comorbidities. Considering the fast-growing elderly population, it is imperative to understand the anatomic, physiologic, and molecular changes of the aging skin and the mechanisms involved in their wound healing process to prevent complications associated with burn wounds. Various studies have shown that stem cell-based therapies improve the rate and quality of wound healing and skin regeneration; however, the focus is on the younger population. In this paper, we start with an anatomical, physiological and molecular dissection of the elderly skin to understand why wound healing is delayed. We then review the potential use of stem cells in elderly burn wounds, as well as the mechanisms by which mesenchymal stem cell (MSCs)-based therapies may impact burn wound healing in the elderly. MSCs improve burn wound healing by stimulating and augmenting growth factor secretion and cell proliferation, and by modulating the impaired elderly immune response. MSCs can be used to expedite healing in superficial partial thickness burns and donor site wounds, improve graft take and prevent graft breakdown.

## 1. Introduction

In 2010, burn injuries cost the Canadian healthcare system $366 million, making it the top 10 in overall spending [1]. While in the U.S., the average cost of burn management for patients is $98,000 for survivors and $310,000 for non-survivors [2].

Burn injuries are extremely distressing, leading to lengthy rehabilitation, permanent disabilities and morbidities that affect the individual’s activities of daily living [3,4]. Wounds from burn injuries are very different from other traumatic wounds, such as lacerations, avulsions, bite wounds, because heat disrupts the body’s homeostasis. Problems arising from burn injuries can be attributed to an increase in the inflammatory and metabolic response, leading to organ dysfunction and systemic failure; burn wound infection from an impaired skin barrier, providing a favorable environment for bacterial proliferation; and a depressed systemic and humoral immune response, putting the patient at risk of sepsis [5]. Scarring is also inevitable which leaves patients with functional and aesthetic deformities [6].

The severity (depth and size) of the burn in combination with the patient’s age, determines the outcome, risk of complications and survival [7]. Managing elderly burn patients is challenging due to poor wound healing, decreased immunologic responses, comorbidities related to aging, diminished sensation and reduced mobility [8,9]. These factors pose significant challenges to the healthcare system, since the number of elderly burn patients continues to rise as the population continues to age [10]. There were 221,519 burns seen in North America between 2009 and 2018, 17% of which were aged 60 years and older. This is alarming since the percentage of elderly burns has increased from 12% to 17% and it is expected to rise [11].

There have been numerous advances in burn wound management, such as laser doppler imaging of burn wounds which improves burn depth assessment in the early period of burn injury, bioengineered dressings which assist in dermal regeneration and the use of stem cells for wound healing. However, most of these studies focus on the younger population which may be due to the higher number of burn injuries in these populations.

Mesenchymal stem cells (MSCs) have been gaining momentum in the field of regenerative medicine and wound healing since 2000 [12,13]. MSCs have been documented to facilitate cell migration, granulation tissue formation, re-epithelialization, angiogenesis and neovascularization, which enhances wound healing [14]. There are currently six ongoing clinical trials on the use of mesenchymal stem cells in burn wounds, all of which are on younger adults [15,16,17,18,19,20].

In this paper, we first describe the anatomical composition and cellular biology of the elderly skin that contribute to impaired burn wound healing. We then discuss mechanisms by which MSC-based therapies may impact burn wound healing in the elderly.

## 2. Elderly Skin

The skin is the largest organ of the body with important physiological functions and acts as a protective barrier. It has an essential role in regulating body temperature, conducting sensory impulses, fat and metabolic products [21]. It also maintains water, electrolyte homeostasis and performs various immunologic functions. It protects the body against U.V. radiation, environmental hazards and infection [22].

The skin’s integrity and functions change as a person ages which leads to compromised healing [23]. The change in the quality of elderly skin occurs as a result of exogenous and endogenous factors [22]. Exogenous changes occur as a result of lifelong environmental exposures, e.g., smoking leading to cumulative DNA damage; U.V. exposure causing daily multiple cellular DNA changes [24]. Endogenous changes are mostly secondary to genetics which is partly responsible for the chronological physiological aging [25,26]. Other factors can be secondary to comorbidities, such as diabetes and vascular diseases which are commonly seen in the elderly population and play a role in delayed healing.

Elderly patients have different skin anatomical compositions and cellular biology compared to the younger population which leads to an intricate interplay of factors, leading to poor wound healing.

### 2.1. Anatomical Composition of the Elderly Skin

The epidermis of elderly skin has a reduced keratinocyte turnover rate, higher pH and decreased water content in the stratum corneum [22,26,27,28]. The higher pH results in a decrease in lipid concentration, leading to defects in the permeability barrier and integrity of the epidermis. These changes in the elderly are visible in the coarseness and irregular pigments of the skin [29,30]. The dermo-epidermal junction rete ridges and dermal papillae are flattened. The surface area for communication between the layers is significantly reduced, leading to a decrease in the flow of nutrients and oxygen [21,31]. Changes in the epidermis make the skin atrophied, wrinkled and less resistant to shearing forces, thus more vulnerable to injury [32]. Shear force is generated by the motion of the subcutaneous tissue and bone relative to an immobilized skin [33].

The dermis of elderly skin has decreased numbers of fibroblasts, mast cells, melanocytes, Langerhans cells, dendritic cells and dermal appendages [34,35,36]. There is also an increased degradation of elastin [37]. Collagen quality is reduced, degradation is increased and a disordered collagen network is increased [38]. These changes lead to the thinning out of the extracellular matrix (ECM). Moreover, dermal vascularity is diminished, resulting in reduced blood flow, impaired nutrient exchange and impaired thermoregulation [39,40]. These changes render the skin less firm, less elastic and more fragile [41] (Figure 1).

### 2.2. Cell Biology of Elderly Skin

On a cellular level, the elderly skin is different as compared to the younger skin which increases the risk of delayed healing and other complications in burn injuries [42]. As the cell ages, gradual decline, translational defects and oxidative stress increase the amount of cellular destruction, while the ability to repair and control DNA damage decreases.

#### 2.2.1. Oxidative Stress

Cells and tissues are generally exposed to low, non-toxic levels of free radicals, such as reactive oxygen species (ROS) and reactive nitrogen species (RNS) [43,44,45]. ROS are produced in the mitochondria as side products in the electron transport chain of glycolysis [38,46]. There is a delicate balance between oxidative stress and antioxidant production [43,47]. This balance means that cells can protect themselves from the harmful effects of free radicals in the normal physiologic condition by several antioxidants and their repair enzymes [48]. This balance is disrupted with aging [49,50]. Cell metabolism is disrupted, leading to mitochondrial breakdown. The defective mitochondria produce excess ROS, causing DNA damage [51,52]. High ROS levels also activate stress signaling pathways such as p38 and Forkhead box protein (FOXO), decreasing the cell’s ability to protect itself from oxidative stress [53]. Oxidative stress with DNA damage and lipid peroxidation increases cell senescence and apoptosis which is deleterious in wound healing, leading to chronic wounds.

#### 2.2.2. Telomere Shortening

ROS and RNS induce changes in skin proteins, resulting in progressive telomere shortening [54,55,56,57,58]. Telomeres are repeating DNA sequences located at the ends of the chromosomes. They cover and protect chromosomal ends from degradation and anomalous recombination [38,59]. They become shorter with each cell division and with aging, ultimately resulting in cellular senescence or further cell death, causing inadequate wound healing [60,61].

#### 2.2.3. Cellular Senescence

Oxidative stress, telomere shortening and DNA damage result in cellular senescence which is the loss of the proliferative capacity or a state of permanent growth arrest [62]. Aging stem cells are affected by senescence through senescence-associated secretory phenotype (SASP) which releases pro-inflammatory signals, thereby disrupting the skin stem cell environment and influencing other cells around the area to deteriorate [62,63]. This contributes to the decline of the regenerative potential of skin tissues [64].

In younger individuals, senescent cells accumulate transiently during the proliferative phase of wound healing to facilitate myofibroblast migration and differentiation [65]. Aged and defective cells, however, accumulate and persist in the elderly, secondary to a failing immune response. The persistent accumulation of senescent cells prolongs the phases of wound healing and delays re-epithelialization [66] (Figure 2).

#### 2.2.4. Immunosenescence

Immunosenescence, the deterioration in cell-mediated immune functions, is associated with aging. Immunosenescence affects various cell types (macrophage, neutrophils, monocytes, fibroblasts, Langerhans cells, T-cells, B-cells, natural killer cells, dendritic cells), as well as reduces the humoral immune responses [42,67,68]. Elderly individuals have a marked increase in the systemic levels of pro-inflammatory cytokines in the absence of insult resulting in a chronic inflammatory state [69,70]. The chronic inflammatory state, combined with increased oxidative stress, creates a vicious cycle which ultimately leads to an increased risk of age-related complications and death [68,71,72].

When exogenous or endogenous factors such as burn afflict an elderly person, the commencing stress response releases more free radicals, initiating lipid peroxidation and high oxidative stress [50,73,74,75,76]. Furthermore, age-related damage to the endothelial cells, prolonged recruitment of leukocytes to the wound and delayed keratinocyte migration and epithelialization confound wound healing [77]. Together, these factors lead to cell death and additional insults on defective elderly skin stem cells, enhancing ischemic tissue necrosis [43,78,79,80,81].

All in all, atrophied, fragile skin, decreased cell repair, inadequate immune response, immunosenescence and pre-existing medical conditions, amongst other problems with aging, predisposes elderly burn patients to delayed healing, higher risk of infection, higher morbidity and mortality [82,83].

The regenerative potential of stem cells might hold the key to address the problem of wound healing in elderly burn patients which is discussed further.

## 3. Stem Cells

### 3.1. Stem Cell Classification

Stem cells are cells that hold the potential to differentiate into many cell types which is contingent on how specialized or potent they already are [84,85]. Following this, stem cells can be classified based on their range of differentiation potential. Totipotent stem cells can differentiate into all types of cells. These can form embryonic and extra-embryonic structures such as the placenta [85]. Pluripotent stem cells, such as embryonic stem cells, can form somatic and germ line cells [86]. Multipotent stem cells can differentiate into specific cell types, in particular tissues or organs; these include hematopoietic stem cells (HSCs), endothelial progenitor cells (EPCs), and mesenchymal stem cells (MSCs) [87]. Stem cells have been used vastly in regenerative medicine due to their unique ability to differentiate into different cell types [88]. However, various ethical issues surround totipotent and pluripotent stem cells, as a result of which MSCs have become popular for research use.

### 3.2. MSC Applications in Regenerative Medicine and Wound Healing

Multipotent MSCs are adult stem cells that can differentiate into mesodermal lineages, including osteocytes, adipocytes, and chondrocytes, as well as neurocytes and hepatocytes, which are of ectodermal and endodermal lineages, respectively [89]. They are defined by the expression of CD105, CD90, CD73 markers, while they lack CD34, CD45, CD11b, CD19, and HLA-DR [90]. The isolation of MSCs has been described from bone marrow, adipose tissue, placenta, amniotic fluid, umbilical cord and skin [89]. Viable MSCs have also been isolated from debrided burned skin that is typically discarded during surgical excision [91]. The availability and differentiation capacity of MSCs have made them excellent candidates for clinical applications.

MSCs have been studied in the context of bone and cartilage reconstruction and nervous system rebuilding, as well as cardiac and liver regeneration [92]. In conjunction with this, research has extended into their applications for wound healing. MSCs promote angiogenesis, re-epithelialization and wound healing, as well as facilitating the reduction of inflammation, making them useful candidates in clinical applications to burn wounds [93]. As a result, these could prove useful in counteracting the negative effects of inadequate healing potential, low collagen quality and diminished dermal vascular density in elderly burns.

## 4. Stem Cells and Wound Repair

Stem cells can facilitate wound healing and closure in burn wounds by promoting ECM reformation, neovascularization, and keratinization [94]. As seen in Figure 3, MSCs can differentiate into various cell types at the site of a wound, as well as recruit fibroblasts, keratinocytes, macrophages and progenitor cells [95]. In addition to this, they also facilitate angiogenesis and neovascularization, as well as influence the growth factor and cytokine profile to promote wound healing [94,95].

### 4.1. MSC Migration and Differentiation

MSCs can render their effects in many ways, one of which is by mobilizing to the site of injury. Evidence for this was seen in two separate mouse wound models where the migration of bone-marrow-derived MSCs (BM-MSCs) to the wound site was tracked after intravenous administration [96,97]. This type of MSC migration and recruitment is based on their diverse chemokine receptor expression profile that promotes chemotaxis [98].

Furthermore, MSCs express and secrete paracrine factors that can activate repair and regeneration mechanisms and enhance cell survival. This has been established through studies using both MSCs as well as MSC-conditioned media [99]. Wharton’s Jelly-derived MSC conditioned media has been shown to upregulate wound healing genes in skin fibroblasts in vitro as well as enhance their proliferation [100]. In line with these findings, BM-MSCs have also been reported to regulate dermal fibroblast proliferation and chemotaxis [101]. The paracrine effects have also been provided by in vivo results, wherein the administration of human adipose-derived stem cells (ASCs) into the sub eschar of mice with full-thickness burns wounds has resulted in the detection of mouse PPARγ and FABP-4, markers of adipogenesis and mature adipocytes, respectively, in the wound bed. It was proposed that human ASCs were exerting their effect by either recruiting mouse progenitors to the wound bed or inducing *de novo* adipogenesis from mouse progenitors [102]. Additionally, MSCs can secrete cytokines and growth factors, promote ECM rebuilding, angiogenesis, neovascularization and modulate the immune response [103].

Although not much is known about the MSCs’ capability to promote healing through direct differentiation into different cell types at the wound site, there is some data available to support this. MSCs have been shown to have the capacity to differentiate into keratinocytes, endothelial cells and pericytes in mouse wound models [97].

Overall, these studies provide evidence of the capability of MSCs to not only mobilize to the site of injury, but also elicit their functions in the wound healing process through paracrine mechanisms and differentiation. Thus, MSCs have a potential use in the elderly to improve senescent skin cells’ function and regenerate damaged tissues. These effects are depicted in Figure 3.

### 4.2. MSCs and Collagen Deposition

Stem cells can also contribute to ECM formation in wound repair through collagen deposition. Although Type I collagen typically exceeds Type III collagen in healthy skin, the remodeling of damaged skin initially depends on Type III collagen deposition [104]. ASCs can promote Type III collagen deposition in a model of rat burn wounds [104]. Additionally, bone-marrow-derived HSCs and MSCs contribute to the long-term collagen deposition during wound healing as they produce collagen Type III and I [105]. Furthermore, mice treated with BM-MSCs show a marked increase in the levels of the vascular endothelial growth factor (VEGF), a cytokine known to play a role in re-epithelialization and collagen deposition [96].

Interestingly, MSCs have also been proposed to play a role in mediating collagen deposition to prevent skin fibrosis which plays a role in scarring. In a model of bleomycin-induced dermal fibrosis, the administration of BM-MSCs resulted in the downregulation of heat shock protein 47 (HSP47) which is implicated in fibrosis [106].

### 4.3. Angiogenesis and Vascularization

MSCs also support wound healing through the promotion of vascularization/vasculogenesis (formation of new blood vessels) and angiogenesis (sprouting of pre-existing blood vessels) [107]. As mentioned earlier, MSCs’ ability to regulate growth factors and cytokines can facilitate vascularization and angiogenesis [94]. For instance, placental MSCs have been shown to release proangiogenic molecules, such as VEGF, HGF, bFGF, TGF-β, and IGF-1 at bioactive levels in culture [108]. Furthermore, ASC-treated mice showed a significant increase in CD-31 expression at burn sites, indicating increased vascularization [102]. Additionally, the BM-MSC treatment of mice with burn injuries resulted in the elevation of VEGF and TGF-β1, both of which are implicated in enhancing neovascularization [96]. These results were also echoed in rats with 30% total body surface area (TBSA) wounds, where human umbilical cord MSCs (UC-MSCs) transplantation increased VEGF levels [109]. The direct differentiation of MSCs has also been proposed to promote vascularization. Placental MSCs can gain endothelial-like morphology and express endothelial markers upon being cultured in endothelial-conditioned media. Direct *de novo* differentiation and paracrine mechanisms of such placental MSCs were proposed to be the causes of increased microvessel density in diabetic rat wounds [108]. Additionally, the endothelial transdifferentiation of MSCs at mouse wound sites were proposed to contribute to wound closures [97].

### 4.4. Immunomodulatory Effects of Stem Cells

Severe burns result in the release of several inflammatory cytokines which can lead to systemic inflammation [110]. They are associated with hypercatabolism and protein wasting which can progress to multi-organ failure, sepsis and even death, especially in the elderly population [94,111,112]. Some of these markers include tumor necrosis factor (TNF)-α, IFN-γ, IL-1β, IL-6, IL-8, IL-12, and IL-17 [94,95,111,113]. Modulating the inflammatory response is an important aspect of caring for burn patients. MSCs can attenuate this inflammatory response by regulating the cytokine profile post-burns, as well as controlling the cell population at the burn site.

Rats transplanted with UC-MSCs post-burn have a faster healing time which is accompanied by lower levels of TNF-α and IL-6 [114]. Similarly, in another study, rats transplanted with UC-MSCs post-burn experienced inflammation to a lesser extent and had lesser macrophage and neutrophil infiltration at the burn site. This was accompanied by a decrease in IL-1, IL-6, and TNF-α and an increase in the anti-inflammatory cytokines IL-10 and TSG-6 [109]. Moreover, the administration of UC- MSC-derived exosomes in rats with third-degree burns decreased the level of TNF-α and lL-1β which were elevated post-burn. There were also increased levels of IL-10 and decreased inflammatory neutrophil and macrophage content [115]. These results were confirmed in cell culture studies, wherein MSCs exposed to inflammatory burn-derived serum upregulated IL-10 expression [116]. Taken together, these findings are interesting, as they provide evidence in favor of MSCs being able to influence the expression of inflammatory cells and regulate the cytokine profile during injuries, especially in the elderly population where immune response is already impaired.

Furthermore, MSCs have also shown immunosuppressive potential in allografts. Pigs that received hemifacial allotransplants demonstrated increased allograft survival and delayed rejection with the administration of MSCs and Cyclosporin A. This group also showed decreased TNF-α and increased IL-10 in circulation two weeks post-transplantation. They also demonstrated a low population of IL-6 positive cells, and a significant increase in the CD25+ regulatory T-cell population [117]. Similarly, prolonged graft survival was seen in baboons that received skin grafts along with intravenous MSC administration [118]. Lastly, the injection of allogenic donor adipose tissue derived MSCs increased graft survival by an additional week in mice that received allogenic skin grafts [119].

These findings highlight the capability of MSCs to suppress allograft rejection by regulating the cytokine profile as well as the cell population. Studies like these are promising for the potential to pursue skin allografts with stem cells where the patient’s own skin is not available for autografts due to poor healing or extensive burns and in elderly patients who are not good candidates for surgery due to their co-morbidities.

## 5. Prospective on Elderly Burns

### 5.1. Stem Cell Role in Repairing Elderly Burn Wounds

Superficial partial-thickness burns typically heal in 7–21 days without functional impairment or hypertrophic scarring [120]. The normal phases of wound healing are prolonged in the elderly [121]. Furthermore, individuals are in a chronic low-grade inflammatory state where inflammatory cytokines are increased 4x than in the younger population even in the absence of infection or other diseases. Thus, their inflammatory response to burn injury is impaired, putting them at risk of various complications [68,69,70]. MSCs can modulate the inflammation by decreasing the circulating cytokines and other products of the inflammatory response in the elderly.

MSCs may also play a role in accelerating wound healing in the elderly by augmenting epidermal and dermal regeneration, by facilitating keratinocyte proliferation and migration and stimulating the production of dermal fibroblasts and collagen [122]. As described earlier, they also improve tissue circulation by increasing the number of new blood vessels [123,124,125]. Overall, MSCs lead to improved wound tensile strength, scar quality and tissue oxygenation. Furthermore, MSCs can influence and accelerate the already delayed phases of wound healing in the elderly and prevent it from becoming a deeper or a chronic non-healing wound by decreasing oxidative stress, activation of essential signaling pathways and attenuating the inflammatory response [126,127,128]. MSCs can inhibit the proliferation of T cells, TNF α, IFNγ and modulate cytokine secretion during the inflammatory phase of wound healing in the elderly. During the proliferative phase, MSCs augment keratinocyte, endothelial cell, fibroblast recruitment and differentiation and facilitate GF secretion. They also improve the collagen quality in the remodeling phase of wound healing in the elderly. This is shown in Figure 4.

BM-MSCs have been reported to successfully heal chronic wound ulcers of the lower extremities [129]. Alternatively, ASCs’ potential to treat burn wounds and ulcers is being studied as they can be easily harvested as compared to BM-MSC [130]. UC-MSCs have been studied in animal models. Allogeneic UC-MSCs have shown low immunogenicity, in vitro and in a mouse model. IFN-γ and TNF-α stimulation did not induce HLA-DR expression in these MSCs [131]. This low immunogenicity of cord-lining membrane MSCs (CL-MSCs) made it a remarkable candidate for clinical use which was recently described in a case report. CL-MSCs were applied topically using a fibrin sealant spray, followed by injection with commercially available MSCs on a patient with >= 70% TBSA that were mostly full-thickness. In addition to faster healing, no hypertrophic scarring or keloids were observed [132]. The ability to isolate viable MSCs from debrided burned skin is inspiring as it would overcome the need for harvesting the patient’s own skin. Immunological reaction and rejection will be negligible since MSCs will come from the patient. Wound healing has been noted without adverse effects in mice and porcine wound models [91]. Studies as such provide evidence for the use of allogeneic MSCs for elderly burns, due to their regenerative capacity, low immunogenicity and lack of rejection potential [132].

### 5.2. Skin Grafts and Donor Site Wound Repair in the Elderly Using Stem Cells

Poor graft take, graft breakdown and delayed donor site healing are significant problems in elderly burns, putting patients at risk of infection.

Early burn wound excision and skin grafting is the standard in managing deep burn wounds. Skin grafting involves removing healthy skin from one area of the body and transplanting it onto a wound [133]. This can be challenging in the elderly, due to their impaired wound healing, poor microvasculature and high susceptibility to infection. Furthermore, harvesting the patient’s own skin may lead to an unintentional new full-thickness wound [134]. The potential role of MSCs in skin grafts is seen in a Phase I Clinical Trial (NCT02104713), wherein BM-MSCs therapy was used on patients >18 years old with deep burns, followed by the application of a split-thickness skin graft. The initial results showed improved graft take and skin regeneration, limited scarring and contracture [135]. The results show the potential role of MSCs in the elderly to enhance the results of split-thickness skin grafting.

Donor site wounds continue to be an area of concern in elderly burns, which can be attributed to a thin dermis and longer healing time of harvest sites [136]. Donor site wounds are the wounds made when the patient’s own healthy skin is removed for split thickness skin grafting. MSCs have the potential to bypass the considerable morbidity associated with donor sites in the elderly by influencing and accelerating the activation, migration and proliferation of different cells involved in the normal phases of wound healing, thereby improving re-epithelialization [117,118,119].

Various skin substitutes have been explored as alternative strategies to circumvent the use of skin grafts and avoid donor site complications. Examples of skin substitutes range from artificial substitutes, human-derived substitutes to porcine substitutes [137]. The types of skin substitutes that are being studied and used range from dermal substitutes and epidermal substitutes to dermal-epidermal substitutes [138].

The use of MSCs with skin substitutes may have a future role as an alternative to skin grafts [139,140]. In an in vitro model, human umbilical vein endothelial cells (HUVECs), ASCs, and fibroblasts seeded onto a fibrin matrix were able to promote a capillary-like structure in which cells were positive for endothelial markers [141]. Furthermore, skin substitutes have also been tested in vivo. Integra^®^ that is transplanted onto mouse wounds along with MSCs, showed significant vascular density [142]. Porcine skin substitutes soaked with MSCs and treated with fibroblast growth factors were able to accelerate wound healing and re-epithelialization in a rat wound model [143]. ASCs or human fetal fibroblasts seeded onto a human amniotic membrane were also capable of lowering inflammation when transplanted onto rat burns [139]. Methodologies as such, have also been applied in clinical settings, wherein MSCs seeded onto an artificial dermis were used to treat wounds [140]. The additional benefit of using skin substitutes is that it eliminates the need for harvesting donor skin, thus avoiding the consequences associated with it.

## 6. Conclusions

As the elderly population continues to increase, so are the cases of elderly burn patients. An increase in the number of elderly burns is significant since wound healing in the elderly is delayed, leading to morbidities, prolonged hospital stays and mortalities. Important changes in the aging skin render it more susceptible to deeper burns, ineffective wound healing and inadequate immunologic response. Based on their relatively easy isolation, potential for proliferation and differentiation and lack of immunogenicity MSCs may play a role in improving burn wound healing in the elderly. They facilitate healing in the already delayed phases of wound healing, modify the impaired immune response, stimulate and augment growth factor secretion and cell proliferation. The use of MSCs has an important prospective role in the elderly population. They can be used to accelerate healing in superficial partial thickness burns and donor site wounds, improve graft take and prevent graft breakdown. This can lead to an overall better quality of life, shorter hospitalization and decreased cost of care.

Further studies are needed to determine the optimal routes of stem cell administration, optimal cell dose to show definitive effectiveness and the fate of the cells after administration, particularly in the elderly.

## Figures and Tables

**Figure 1 ijms-21-04604-f001:**
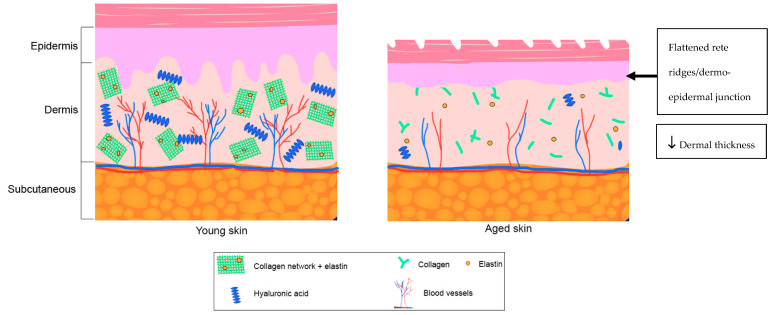
Differences in skin structure between young and aged skin. Collagen and elastin in the aged skin has a reduced quality and a disorganized network. Hyaluronic acid is decreased and vascularization is decreased.

**Figure 2 ijms-21-04604-f002:**
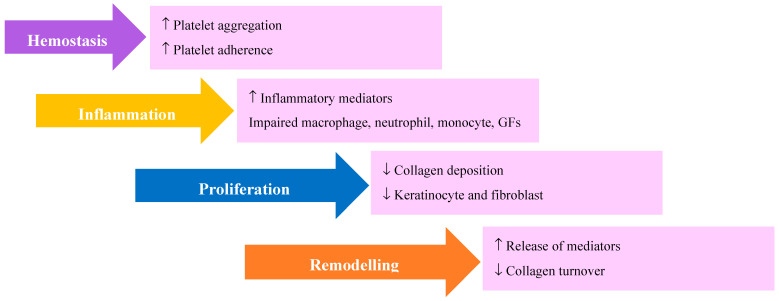
Phases of wound healing with age-related changes. Hemostasis - there is increase in platelet aggregation and adherence. Inflammation –there is increase in inflammatory mediators along with impaired macrophage, neutrophil, monocyte and GFs. Proliferation – there is decrease in keratinocyte, fibroblast and collagen deposition. Remodelling – there is increase in release of inflammatory mediators and decrease collagen turnover. Inflammatory mediators are interleukin (IL)-1β, interferon-γ (IFN-γ), IL-1 receptor antagonist (RA), IL-6, granulocyte macrophage colony-stimulating factor (GM-CSF), and FMS-like tyrosine kinase 3 ligand (FLT-3L). Growth factors are platelet derived growth factor (PDGF), transforming growth factor (TGF), fibroblast growth factor (FGF), epidermal growth factor (EGF) vascular endothelial growth factor (VEGF), keratinocyte growth factor (KGF).

**Figure 3 ijms-21-04604-f003:**
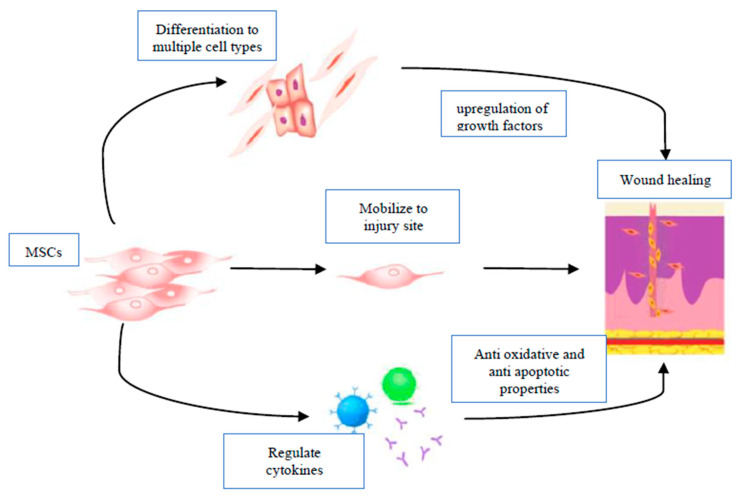
Mesenchymal stem cell (MSCs) mechanisms for wound repair. MSCs may facilitate wound healing through mobilization to wound site and differentiation to keratinocytes, dermal fibroblasts and other cell types. They also regulate the release of pro-inflammatory cytokines (IL-1β, IFN-γ, IL-1RA, GM-CSF, and FLT-3L) and stimulate the release of growth factors (PDGF, TGF, FGF, EGF, KGF, VEGF).

**Figure 4 ijms-21-04604-f004:**
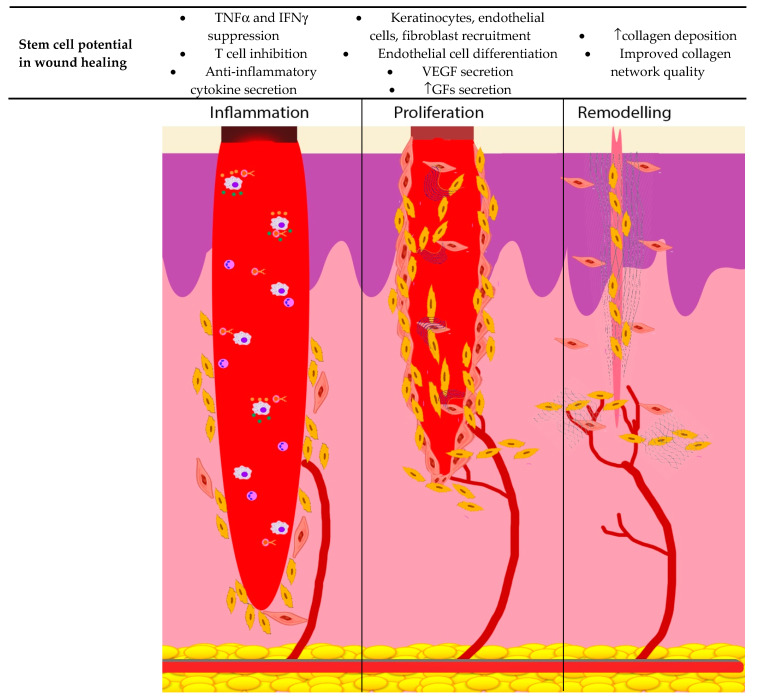
MSCs roles in each phase of the wound-healing process. Inflammatory phase, MSCs inhibit the proliferation of T cells, TNF α, IFNγ and modulate cytokine secretion. Proliferative phase, MSCs augment keratinocyte, endothelial cell, fibroblast recruitment and differentiation and facilitate GF secretion. Remodelling phase MSCs improve collagen quality.

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
