# Peer review of "Burns in the Elderly: Potential Role of Stem Cells"

_ijms, 2020, doi:10.3390/ijms21134604_

Round 1
Reviewer 1 Report
Dear Authors,
congratulation for your review. I found it interesting, innovative and well-structured. The prospective use of MSCs for the treatment of burns is very actual and the review emphasizes all the possible benefits of this treatment for elderly skin. Below my comments:
Lines 78-79: To specify that the differences of skin anatomical composition and cellular biology are between elderly and younger patients
Lines 91-93: Are the authors sure that there is a decreased degradation of elastin? To insert references for sentences regarding elastin and collagen.
Figure 1: It is better to report below the figure 1 the various components of the figure related to young and aged skin with a legend to indicate what they correspond to.
For example: hyaluronic acid in young (image) and aged skin (image)
The various changes in the elderly skin can be indicated in a separate scheme without arrows.
The change in the elderly skin regarding the flattened rete ridges/dermo-epidermal junction can be maintained in the right side of the figure 1 with arrow.
The change in the elderly skin regarding decreased dermal-thickness can be maintained in the same actual position
Replace younger skin with young skin and older skin with aged skin as described in the figure legend
Figure 4: the bulleted list is enough to indicate the various points (remove the arrows)
Lines 284, 322, 332: Donor skin seems to refer to the use of homologous skin. If you can, change it.
Lines 333-337: Have the results been obtained on elderly or young patient? Please, specify it.
Line 371: To emphasize the important role of MSCs and their prospective clinical use for burn treatment in conclusions
Author Response
Thank you very much for your comments and suggestions.
Lines 78-79: I added a comparison to the younger population.
Lines 91-93: Thank you very much for this. There's increased degradation of elastin. References indicated for elastin and collagen.
Figure 1: Made the necessary changes and added figure legends.
Figure 4: Arrows removed.
Line 284, 322, 332: Donor skin is the patient's own skin, which I replaced.
Lines 333-337: The study recruited burn patients >18 years old. They did not specify the age. I sent the authors an email, I'm hoping they will get back to me.
Line 371: Emphasized on the elderly population and the prospective role of MSCs.

Reviewer 2 Report
The review article entitled “Burns in the elderly: Potential role of stem cells” by Margarita Elloso et. al, presents the state-of-the art of burns wound healing, with emphasis of impaired regenerative function of the skin in the elderly patients. The authors discuss the utility of mesenchymal stem cell-based therapies in wound healing, and how elderly patients can benefit from such therapy. The article is written very clearly and the data described are supported by numerous important references, including the data from in vivo experiments. The text is supplemented with four figures, which help to better understand the discussed data. I find the article very well prepared. The only one thing I would suggest to improve, is to list directly in the figures, the specific cytokines, growth factors and other secreted molecules, which participate in the processes shown in the figures 2 and 3. This will make the figures more comprehensive and easier to read.
Author Response
Thank you very much for the comments. I have made the necessary changes on figures 2 and 3.

Reviewer 3 Report
The manuscript provides a nice overview of studies using MSCs and their effect in the treatment of burns with a focus on the elderly population. Some discussion about the possibility of autologous versus allogeneic application and their advantages and disadvantages would be beneficial.
Author Response
Thank you very much for your comments. Unfortunately, there are not a lot of literature regarding autologous vs allogeneic stem cell application on wounds. Most are case studies. I briefly mentioned the prospective use of autologous burn derived mesenchymal stem cells in 329-331. This will prospectively address the immunologic reaction and rejection of allogeneic sources.
